# Corporate governance, financial performance, and economic policy uncertainty. Evidence from emerging Asian economies

Chaoyang Zhang ⓘ *

School of Accountancy, Henan Institute of Economics and Trade, Zhengzhou, Henan, China

* vhikyh@qq.com

**Data Availability Statement:** The data is attached as support information file.

**Funding:** The author(s) received no specific funding for this work.

## Abstract

This study examines the impact of corporate governance and economic policy uncertainty on financial performance in non-financial firms in three emerging Asian economies: Pakistan, China, and Malaysia. The study analyzed 400 non-financial firms listed on these countries' stock markets for ten years (2012–2021). Data on corporate governance, financial performance, and CSR scores were obtained from DataStream. The study used regression analysis and the generalized method of moments (GMM) for its robustness analysis. Our findings show that all attributes of corporate governance practices have a significant positive impact on return on assets, except for the existence of an audit board committee for all selected economies. Moreover, corporate governance practices have a significant positive relationship with return on equity. However, in the case of earnings per share (EPS), all attributes have a significant positive relationship except board size with earnings per share. Economic policy uncertainty significantly moderates corporate governance practices and financial performance of organizations belonging to Asian economies. This study advocated the implications for the government and policymakers to improve corporate governance practices, especially during periods of high economic uncertainty.

## 1. Introduction

The corporation's financial performance plays a vital role in the development of the countries with the rise to the income level of the shareholders because a company's financial health and viability can be inferred from financial performance parameters. These parameters assist stakeholders in determining if the business is making enough money to pay its debts and cover its expenses. Shareholders and prospective investors carefully monitor a company's financial results. Positive financial outcomes increase stock price and investor confidence, which makes it simpler for the business to raise funds through equity financing. Moreover, financial performance information is used by creditors and lenders to assess a company's creditworthiness. A corporation gets financing when necessary and obtains advantageous borrowing terms with a solid financial history. However, the financial performance of corporations has a broader economic impact. It contributes to economic growth by creating jobs, generating tax revenue, and fostering innovation.

**Competing interests:** The authors have declared that no competing interests exist.

Similarly, corporate financial performance remains a problem because it is affected by many factors, such as social and economic factors [1]. Strong corporate governance practices are linked to the corporations' success and long-term sustainability and provide a correct direction to the business operations in the country. Therefore, smooth corporate governance practices should be necessary for each firm to develop over time. Moreover, accountability and openness are stressed in strong corporate governance standards regarding financial reporting and decision-making. A company's stakeholders are likelier to have faith and confidence in it when it provides accurate and transparent financial information. This can then draw in investors and possibly cut the cost of capital, resulting in better economic performance [2]. However, investors and stakeholders increasingly consider a company's risk associated with the economic policies when assessing its financial success. A company's reputation and attractiveness to socially conscious investors can be improved by having sound corporate governance policies that address economic policy uncertainty.

Most pertinent research endeavors have delved into corporate governance's primary and frequently prevailing facets, including board size, board characteristics, and other strategic practices that significantly influence corporate financial performance [3]. In this way, firms get more profitability to pick the highly efficient management mode and consider efficient corporate governance practices [4]. There is rich potential in resolving doubts regarding the corporate governance mechanism's broader role, which could benefit firms. However, it is still unresolved due to the wide range of significance with respect to time domain changes and differences in the culture of each economy [5, 6].

Economic policy uncertainty has a wide range of effects on the economy as well as on the financial performance of the organization [7, 8]. According to [9], corporate governance practices have different consequences on financial performance. Moreover, [10] explain the effect of economic policy uncertainty on financial indicators and provide the research gap for further investigation related to the moderating effect of economic policy uncertainty. The previous studies have data limitations on economic policy uncertainty [11], but this study select the gap for investigation with new insight with inclusion of wide range of dataset.

Pakistan's relatively nascent corporate governance framework often struggles with enforcement, which hinder financial performance, particularly during periods of high economic policy uncertainty [12]. Similarly, China with more centralized governance structures and state-owned enterprises, demonstrates a different dynamic where government policies and interventions mitigate some uncertainty but may also stifle market-driven growth [13]. In contrast, Malaysia exhibits a more developed corporate governance system that fosters better financial performance in the face of uncertainty, with robust regulatory frameworks allowing for greater market resilience [14]. The influence of economic policy uncertainty in each country is shaped by the strength of institutions and governance practices, impacting corporate outcomes differently. Therefore, this study extracts a gap in literature to investigate the corporate governance along with the economic policy uncertainty effect the financial performance in different culture that provide strong evidence of both strong and week corporate governance structure. Similarly, this study is also important for improving the corporate sector in the emerging economy where corporate governance mechanism is week.

This study encompasses two primary objectives. First, it aims to scrutinize the relationship between various attributes of corporate governance and corporate financial performance by utilizing diverse measures such as return on assets (ROA), return on equity (ROE), and earnings per share (EPS). The second objective examines the moderating influence of economic policy uncertainty on the relationship between corporate governance and corporate financial performance.

This study makes significant new contributions in multiple dimensions, effectively filling gaps left by previous researchers. First and foremost, it broadens the scope by encompassing 400 top-listed corporations from three distinct stock exchanges, resulting in a much larger sample size. Previous research, such as [15, 16] used small samples from a single country. Third, whereas previous research, such as [9, 17] frequently focused on short periods, this study broadens its scope to include a 10-year sample span. Fourth, unlike previous studies that focused on a single nation as their Sample, such as [15, 16], this research broadens the reach to include enterprises from three different Asian countries. Fifth, this study distinguishes itself by applying advanced econometric methods, such as composite effects that consider all listed enterprises in all selected nations. Furthermore, in contrast to previous investigations, such as [9], which relied on more straightforward methodologies, separate country-specific effects were studied using dummy analysis. Finally, to obtain a more accurate result, economic policy uncertainty is used as a moderator that has a positive moderating effect.

This study underscores the significance of solid governance frameworks in reducing uncertainty and improving financial performance. This study provides policy implications for the government to implement policies encouraging openness, accountability, and regulatory stability to boost investor trust and attract investment. Investors should prioritize due diligence on corporate governance procedures when making investment decisions, considering the influence on financial returns. Legislators are encouraged to create legislation that enforces robust governance standards and provides efficient dispute-resolution processes, fostering a favorable climate for the region's long-term economic growth and development.

The rest of the article is organized as follows: Literature review section provides an overview of research theoretical underpinnings, giving many testable hypotheses to understand variable interactions. The methods section describes the conceptual framework, research design, and methodology. In the sext section, study findings and discussion are explained. At the end conclusion, future research suggestions, limitations, practical implications, and policy implications are illustrated.

## 2. Literature review

### 2.1 Theoretical background

Every research study is grounded in foundational theories. This study draws upon agency, stewardship, stakeholder, and resource dependence theories as its theoretical underpinning theories [18–20]. According to these foundational theories, corporate governance is critical in resolving conflicts between management and shareholders, aligning management objectives with the interests of investors, and maximizing corporate finances [5, 21]. These theories also advocate for management to act in the best interests of both shareholders and stakeholders. Similarly, according to [22], the board size is important due to the decision-making process. The larger board size creates a conflict of timely decision-making by the board that has greater chances of agency conflicts. [23] explained that the existence of audit committees increases the organizations' financial performance by checking the internal control with respect to time.

This study is also justified by stakeholder theory, which emphasizes prioritizing stakeholders through protecting their rights and producing long-term value through good corporate governance processes and CSR disclosures. According to [24], the existence of the CSR committee creates investor confidence in the organizations, which leads to the rise of foreign direct investment and financial performance. Moreover, the auditor independence and the audit committee expertise increase the financial performance by increasing the investors' confidence in the firms [25].

Numerous studies have explored the relationship between corporate governance and corporate financial performance and have produced a body of evidence that supports the notion that effective corporate governance practices significantly influence a firm's financial and operational performance such as [26] defines the same relationship and concluded the positive relationship between them. Furthermore, [27] investigates the effect of corporate governance on firm performance, notably in the context of the COVID-19 epidemic. Their findings show that companies with strong corporate governance measures, including board independence and CEO duality, fared better during the pandemic and maintained their financial performance. Notably, good corporate governance emphasizes the significance of risk management. Firms with well-established risk management systems are better positioned to recognize, assess, and mitigate various financial, operational, legal, and reputational risks. This astute risk management strategy aims to avoid potential financial losses and protect the company's assets, strengthening its financial performance [28].

## 2.2 Governance practices and corporate financial performance

Corporate governance practices and the financial performance of firms are widely discussed in the literature. Corporate governance practices vary across the country depending on the environment and culture of the economy. Many countries have family-owned firms, but others may be public-owned firms. According to [29], efficient corporate governance practices increase corporate performance by reducing risk factors, increasing decision efficiency, and helping in decision-making that improves performance directly. The audit committee and the board characteristics that follow the code of corporate governance improve financial performance in terms of increasing return and profitability [30].

[31] also found a significant positive relationship between corporate governance practice and performance and concluded the positive effects of the independent board on performance in China. Moreover, an efficient board improves financial performance by making timely decisions, solving problems, and taking opportunities through protection and decision-making [32].

[33] investigated the positive relationship between board size and performance and concluded that a larger board has a significant sign for organization promotion due to effective decision making and brainstorming that reduces agency problems. Similarly, [34] found the negative consequences of the larger board on the performance due to increased corporation expenses that lead to decreased profitability. However, [35] concluded the positive relationship between larger boards and performance.

[36] found a positive relationship between board diligence attendance and performance. Also, they concluded that the board's attendance at meetings increases the brainstorming and thinking power of the board in making timely decisions. Similarly, [37] investigated the positive effects of the presence of the board of directors on profitability due to the management of top-level decisions. Moreover, [22] stated that the larger board size significantly negatively impacts financial performance due to the conflict in decision-making.

[38] investigated that the audit committee attracts foreign investors due to an increase in the investors' confidence in the firms due to the mitigation of the risk of fraud. [39] explained the positive consequences of the audit committee on profitability due to risk reduction and removing the weakness of internal control. Moreover, [40] define the positive relationship between audit committees and performance. [41] investigated that financial performance is also positively affected by the expertise and independence of the audit committee, attributed to their wealth of experience and impartial perspectives. Based on the theoretical and empirical evidence, we developed the following hypothesis,

*H1*: Board size has a significant positive impact on performance.

*H2:* Board attendance has a significant positive impact on performance.

*H3:* CSR committee existence has a significant positive impact on financial performance.

*H4*: The audit board committee has a significant positive impact on performance.

*H5*: Audit committee independence has a significant positive impact on financial performance.

*H6*: Audit committee expertise has a significant positive impact on financial performance.

## 2.3 Moderating role of economic policy uncertainty

Economic policy uncertainty affects different countries in various ways, depending on their financial structures, political systems, and exposure to global markets. While some nations may experience increased caution and reduced investment during periods of uncertainty, others may respond with adaptive strategies, innovation, and diversification, underscoring the complexity of its global impact.

Economic policy uncertainty notably influences corporate governance practices, as evidenced in recent literature. Scholars have increasingly examined how fluctuations in economic policy uncertainty impact corporate decision-making, risk management, and overall governance strategies. High levels of economic policy uncertainty often led companies to adopt a more cautious approach, potentially affecting their investment decisions, dividend policies, and executive compensation structures [42]. A climate of uncertainty also stimulates a stronger emphasis on risk management within organizations as companies seek to navigate turbulent economic landscapes and safeguard their financial interests [43]. Moreover, [44] suggest that companies facing elevated economic policy uncertainty may pay greater attention to corporate governance mechanisms to mitigate potential risks and maintain investor confidence. These findings underscore the complex interplay between economic policy uncertainty and corporate governance practices, offering valuable insights into how external economic factors can shape internal governance dynamics and, ultimately, impact firm performance and stakeholder interests.

Various literature has extensively investigated the impact of economic policy uncertainty on corporate financial performance. Studies have shown that higher levels of economic policy uncertainty can lead to decreased investment, reduced firm valuations, and increased risk premiums, ultimately negatively influencing financial performance [45]. Firms in uncertain economic environments often become more cautious in their decision-making, leading to delayed investments and a focus on liquidity management to mitigate potential risks [46]. Moreover, economic policy uncertainty can affect firms' access to capital, as lenders and investors may become more risk-averse, leading to higher financing costs [47]. However, it is essential to note that the specific effects of economic policy uncertainty can vary across industries, regions, and firm sizes, reflecting the nuanced relationship between economic policy dynamics and corporate financial performance.

Furthermore, some firms may exhibit adaptability and innovation in response to uncertainty, which can positively influence their financial outcomes, emphasizing the multidimensional nature of this relationship [48]. Hence, on the ground of previous literature discussion, we assume the positive moderating influence of economic policy uncertainty. On the grounds of the discussion, we developed the following hypothesis.

*H7*: Economic policy uncertainty has a significant positive moderating impact between corporate governance attributes and financial performance.

## 3. Research design

### 3.1 Data collection and sample size

To meet our study objectives, this study used the 400 firms' data from three different economies belonging to the Asian region from 2012 to 2021 on annual frequency. Every country has its different corporate governance practices and control of the corporate sector. China has centralized governance structures and state-owned enterprises, Pakistan is struggling to make the corporate control structure in high economic policy uncertainty, and Malaysia exhibits a more developed corporate governance system that fosters better financial performance in the face of uncertainty. Therefore, we selected Pakistan, Malaysia and China that contains different firms' structure, corporate control and facing various type of economic policy uncertainty in their economy. Similarly, this study aimed to provide a significant implication for investor to invest in different firm's portfolio belonging to different corporate structure and control because these three economies provide a different structure of the control. Moreover, these selected economy changes the corporate governance mechanism more frequently according to their economic needs.

The period chosen for analysis (2012–2021) is significant due to the changes made by certain economies regarding specific policies and their financials during this period. Non-financial companies were selected for analysis due to the distinct social, cultural, and regulatory settings that differ from those observed in the financial sector, where similar cultural and regulatory settings are commonplace. The financial sector is excluded because it has separate corporate governance mechanisms and is controlled by two different legislators, such as the Security Exchange Commission and the reserve bank of that economy. Including the financial sector in the same analysis will make biased results.

There are two sources of data collection. First, the data related to corporate governance, Economic policy uncertainty, firm age, and firm size is collected from the DataStream database. Second, the data associated with GDP data for these years are extracted from the World Bank data portal. The sample size from each economy is given below in Table 1. In Pakistan, 750 firms have high market capitalization, China has 1200 firms, and Malaysia has 967 organizations. These organizations show our population in the selected economies. The sampling technique used in the study is probability sampling, which depends on the availability of the data of these firms in the respective economies. Table 1 displays the sample size for each economy. The Sample comprises the higher market capitalized companies from each country selected through random sampling techniques of probability sampling. These companies contribute significantly to the total market capitalization of their respective countries.

Fig 1 represents the conceptual framework of the study is given below.

### 3.2 Variables measurement

**3.2.1 Dependent variable.** The critical dependent variable under consideration in this study is financial performance, measured using several measures. Return on assets (ROA), return on equity (ROE), and earnings per share (EPS) are examples of these indicators.

**Table 1. Number of the firms taken as a sample.**

| Name of the Country | Non-Financial firms Sample |
|---|---|
| Pakistan | 150 |
| China | 127 |
| Malaysia | 123 |

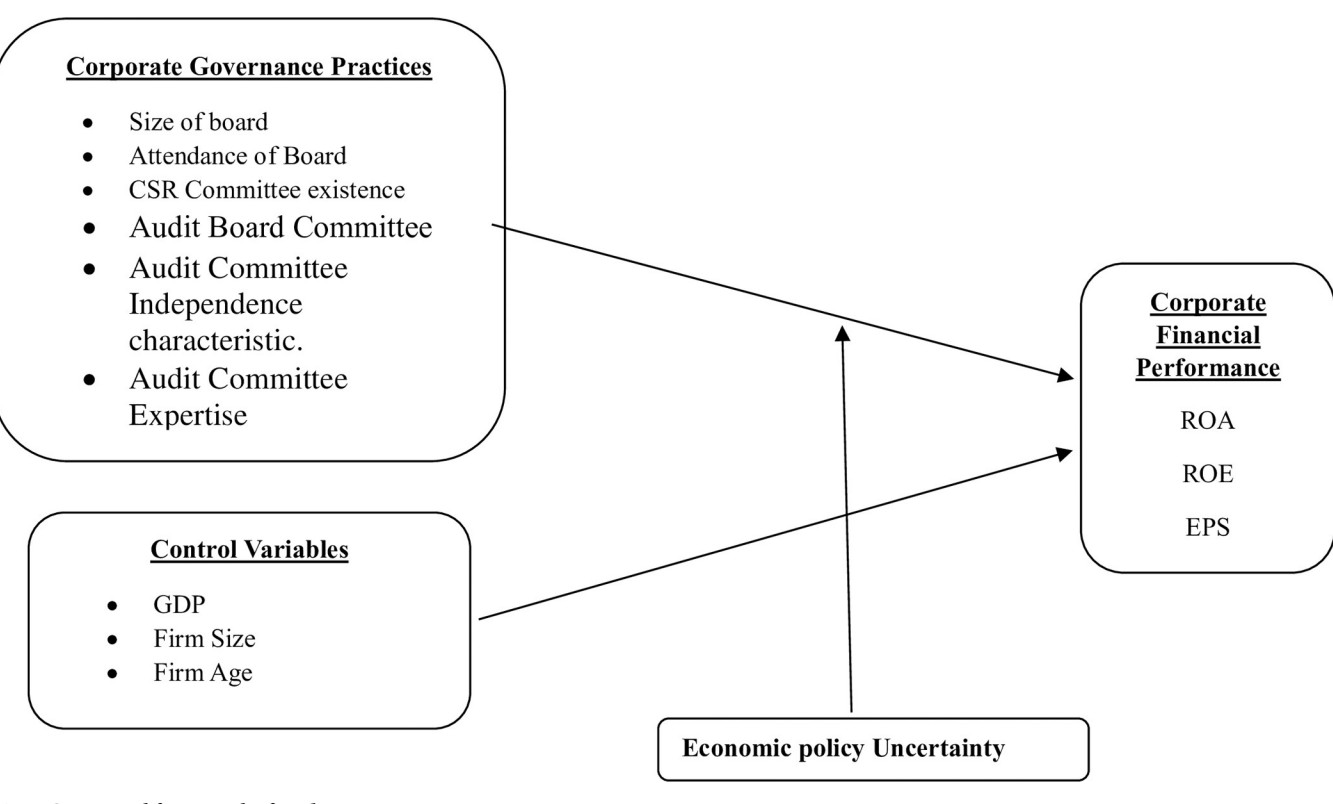

**Fig 1. Conceptual framework of study.**

Among these indicators, return on assets (ROA) is well-known in finance and is frequently used to assess business financial performance. ROA is calculated by dividing a company's net profit by its total assets, and it indicates how well a company leverages its assets to maximize output. The return on assets is considered the most important for measuring financial performance [49].

Furthermore, return on equity (ROE) is a popular metric for assessing business financial performance. It is calculated by dividing net income by total shareholders' equity and reflects the effective use of owner's equity to achieve profitability. The utilization of the equity is essential for generating the profitability [50]. Furthermore, earnings per share (EPS) is a third indicator of financial success. Profitability is a commonly acknowledged metric for evaluating business financial performance [51].

**3.2.2 Independent variable.**   This study's independent variable is corporate governance practices, evaluated based on various criteria, including board and auditor characteristics. Notably, corporate governance practices considered in this study include board size, board attendance, and the inclusion of CSR committee on the board. Furthermore, auditor-related qualities include the existence of an audit board committee, auditor independence, and audit committee expertise. Board size (BS) is measured by the total number of directors in board composition. It also shows the number of directors on the board [25]. Board attendance is measured by the board member's presence in the meeting [52], and board CSR committee is measured through the dummy; if they existed, then denoted by 1, if not 0. The audit board committee is also measured through a dummy; if it exists, it is denoted by 1, and if not, it is denoted by 0. The audit committee expertise score measures the auditor committee's expertise [53]. Audit committee independence (ACI) is measured through the dummy; if they existed, then denoted by 1, if not 0 [51].

**3.2.3 Control variables.** Control factors such as GDP, firm size (FSIZ), and firm age (AGE) are included in this study. Total assets are used to determine business size since they are a vital factor influencing firm performance. GDP, which measures economic output inside a particular country, impacts enterprises' financial success. Meanwhile, firm age is measured in years since the date of incorporation in the stock market, providing insight into the maturity of enterprises relative to total market capitalization.

**3.2.4 Moderator.** The moderator variable is the economic policy uncertainty in this study. Economic policy uncertainty creates an index based on three major components to measure policy-related economic uncertainty [54]. The first and most adjustable component estimates the amount to which policy-related economic uncertainty is covered in the media. The second-ranking component is based on information compiled by the Congressional Budget Office (CBO), which compiles lists of temporary provisions in the federal tax code. The policy-related uncertainty index's third component is drawn from the Federal Reserve Bank of Philadelphia's Survey of Professional Forecasters.

## 3.3 Econometric model and method of estimation

This study empirically examined the impact of corporate governance on firm performance within listed companies operating in three emerging economies within a specific region over ten years. Drawing on pertinent literature, this research employed regression analysis techniques.

$$FP_{it} = \alpha_0 + \beta_1(CG)_{it} + \beta_2(EPU) + \varepsilon_{it}$$

$$FP_{it} = \alpha_0 + \beta_1(CG)_{it} + \beta_2(CG)*EPU_{it} + \varepsilon_{it}$$

Regression analysis is a widely adopted approach for assessing the connection between financial variables. It entails estimating the association between a dependent variable and one or multiple independent variables. In finance, the dependent variable frequently represents a measure of financial performance, such as return on assets or stock price. In contrast, the independent variables encompass factors that could impact the dependent variable, including interest rates, inflation, and firm-specific variables like size or leverage. In finance and economics, regression analysis is a prevalent and valuable statistical tool employed to investigate and understand the relationships between variables. [55] used the regression method to analyze the risk in the context of China. Moreover, [56] investigate the volatility relationship between oil prices and the stock market using the regression analysis method.

Moreover, the regression analysis is weak regarding the simultaneous effect between endogenous and exogenous variables. To remove the simultaneous effect and problem of the autocorrelation (if any) between endogenous and exogenous variables, there should be the use of the generalized method of movement (GMM) that is useful in econometrics. We used the GMM method for real impact on our finding's robustness. Therefore, weak instruments can result in skewed parameter estimations in instrumental variable regression models in econometrics. When instrumental variable approaches are required, GMM is an essential tool since it offers a mechanism to test for and address the issue of weak instruments.

Moreover, GMM is a preferred choice for robustness analysis due to its versatility in handling various types of data and model specifications [57]. Unlike parametric methods that rely on strict distributional assumptions, GMM offers flexibility by allowing researchers to specify moment conditions based on observed data without requiring explicit distributional assumptions. This makes GMM particularly suitable for analyzing complex datasets whose underlying distribution may be unknown or non-standard. Moreover, GMM accommodates

heteroscedasticity and autocorrelation, standard features in real-world data, by providing efficient estimation methods that can robustly capture underlying relationships even in such complexities.

# 4 Results and findings

## 4.1 Descriptive statistics

In Table 2 above, the selected firms exhibit an average return on assets of 33%, indicating their ability to generate, on average, a 33% return on their assets. Moreover, these firms' average return on equity is notably five times higher than their average ROA, underscoring a favorable return on shareholders' equity. The mean earnings per share value stands at 1.13, indicating a consistent positive trend in firm earnings over the past decade. Furthermore, Economic policy uncertainty has a mean value of 13.13, ranging from 0 to 19 among the selected firms in Asian economies. Approximately 51% of firms feature an audit committee, and the firms, on average, have been in operation for 16 years since their incorporation. Furthermore, about 44% of firms have board attendance at board meetings, and the average board size comprises 11 members. Among the Asian economy data, 66% of firms have a CSR committee.

Table 3 indicates that no correlations exceed the 0.80 threshold between variables. When the correlation between variables exceeds 0.80, it may hinder regression analysis. In our case, the absence of such strong correlations allows for a suitable regression analysis. While there is a correlation between ROA and ROE, both are dependent variables. Since only one measure of financial performance is employed, they do not significantly impact the overall regression results.

## 4.2 Results and findings

The regression analysis in Table 4 sheds light on the impact of corporate governance on proxies for company financial performance, specifically in the context of Pakistan, as represented by the country dummy. The R-squared values of 0.7410, 0.7982, and 0.4210 represent significant variations in the dependent variables ROA, ROE, and EPS due to the explanatory variables, which include board-related and audit-related characteristics of corporate governance

**Table 2. Descriptive statistics of the samples.**

|  | Mean | Maximum | Minimum | Std. Dev. | Observations |
|---|---|---|---|---|---|
| ROA | 0.33 | 6.200 | -1.10 | 6.18 | 4000 |
| ROE | 4.84 | 7.280 | -5.95 | 2.48 | 4000 |
| EPS | 1.13 | 2.81 | -12.37 | 6.70 | 4000 |
| EPU | 13.13 | 19.22 | 0.00 | 5.70 | 4000 |
| ABC | 0.85 | 1.00 | 0.00 | 0.36 | 4000 |
| ACE | 51.47 | 100.00 | 0.00 | 32.33 | 4000 |
| ACI | 67.00 | 100.00 | 0.00 | 37.16 | 4000 |
| AGE | 15.99 | 120.00 | 1.00 | 14.97 | 4000 |
| BA | 0.45 | 1.00 | 0.00 | 0.50 | 4000 |
| BS | 11.00 | 19.00 | 7.00 | 7.09 | 4000 |
| CSC | 0.66 | 1.00 | 0.00 | 0.47 | 4000 |

Note: ROA used for return on asset, ROE for return on equity, EPS for Earning per share, EPU for Economic policy uncertainty, ABC for audit board committee, ACE for audit committee expertise, ACI for audit committee independence, AGE for firm age, BA for board attendance, BS for board size and CSC for CSR committee existence.

**Table 3. Correlation matrix.**

|  | ROA | ROE | EPS | EPU | ABC | AC | ACI | AGE | BA | BS | GDP | FS |
|---|---|---|---|---|---|---|---|---|---|---|---|---|
| ROA | 1 |  |  |  |  |  |  |  |  |  |  |  |
| ROE | 0.54 | 1 |  |  |  |  |  |  |  |  |  |  |
| EPS | 0.01 | 0.03 | 1 |  |  |  |  |  |  |  |  |  |
| EPU | -0.03 | -0.03 | 0.08 | 1 |  |  |  |  |  |  |  |  |
| ABC | -0.04 | -0.04 | 0.06 | 0.59 | 1 |  |  |  |  |  |  |  |
| ACE | -0.03 | -0.03 | -0.03 | 0.43 | 0.64 | 1 |  |  |  |  |  |  |
| ACI | -0.03 | -0.03 | -0.06 | 0.47 | 0.66 | 0.55 | 1 |  |  |  |  |  |
| AGE | 0 | -0.01 | -0.02 | -0.05 | -0.04 | -0.02 | -0.05 | 1 |  |  |  |  |
| BA | -0.01 | -0.01 | 0.02 | 0.17 | 0.35 | 0.35 | 0.44 | -0.01 | 1 |  |  |  |
| BS | 0.01 | 0.01 | -0.01 | 0.03 | 0.02 | -0.02 | 0.02 | -0.01 | 0.03 | 1 |  |  |
| CSC | -0.03 | -0.03 | 0.01 | 0.65 | 0.54 | 0.42 | 0.47 | -0.07 | 0.24 | -0.06 |  |  |
| GDP | -0.01 | -0.01 | 0.03 | 0.18 | 0.21 | 0.12 | 0.11 | -0.36 | 0.08 | -0.08 | 1 |  |
| FS | 0.03 | 0.04 | 0.04 | 0.14 | 0.01 | 0.01 | 0.07 | 0.02 | -0.03 | 0.01 | 0.02 | 1 |

Author source

Note: ROA used for return on asset, ROE for return on equity, EPS for Earning per share, EPU for Economic policy uncertainty, ABC for audit board committee, ACE for audit committee expertise, ACI for audit committee independence, AGE for firm age, BA for board attendance, BS for board size and CSC for CSR committee existence

mechanism and are statistically significant at the 1%, 5%, and 10% levels of significance, respectively. Furthermore, the positive F-statistics highlight the model's appropriateness and fitness in explaining the observed outcomes.

**Table 4. Results of regressions.**

| | ROA | | ROE | | EPS | |
|---|---|---|---|---|---|---|
| Variable | Coefficient | Prob. | Coefficient | Prob. | Coefficient | Prob. |
| ABC | 0.677 | 0.134 | 39.541 | 0.076 | 3.871 | 0.000 |
| ACE | 0.001 | 0.017 | 0.001 | 0.097 | 0.020 | 0.000 |
| ACI | 0.002 | 0.018 | 0.070 | 0.009 | 0.033 | 0.000 |
| AGE | 0.007 | 0.090 | 0.178 | 0.006 | 0.009 | 0.155 |
| BA | 0.167 | 0.001 | 2.441 | 0.076 | -0.868 | 0.000 |
| BS | 0.000 | 0.834 | 0.000 | 0.025 | 0.000 | 0.302 |
| CSC | 0.297 | 0.083 | 2.013 | 0.055 | -0.934 | 0.000 |
| EPU | 0.006 | 0.025 | 0.097 | 0.035 | 0.021 | 0.000 |
| FS | -0.001 | 0.008 | 0.013 | 0.044 | 0.008 | 0.071 |
| GDP | 0.000 | 0.667 | 0.000 | 0.017 | 0.000 | 0.919 |
| China | -0.241 | 0.0263 | -7.218 | 0.076 | 0.117 | 0.686 |
| Malaysia | 2.059 | 0.0224 | 29.746 | 0.069 | 0.351 | 0.559 |
| C | 1.775 | 0.0003 | 42.572 | 0.007 | 0.358 | 0.307 |
| R-squared Value | 0.20410 | | 0.0982 | | 0.4210 | |
| Adjusted R | 0.20301 | | 0.0871 | | 0.4014 | |
| Probability | 0.0000 | | 0.0000 | | 0.0000 | |
| Durbin-Watson | 1.0104 | | 1.1701 | | 1.1214 | |

Note. Findings are presented with respect to Pakistan as the country dummy.

The audit board committee (ABC) does not have a statistically significant impact on performance, as indicated by non-compliance with acceptable probability criteria, according to the study of ROA results. Our null hypothesis is confirmed in this finding. However, at the 5% level of significance, the presence and independence of the audit committee have a significant and favorable effect on performance. This means that a one-unit increase in the audit committee's existence and independence corresponds to a 0.001 and 0.002 increase in ROA, respectively. Furthermore, all other components of corporate governance have a significant and positive influence, supporting the view that good governance procedures improve business financial performance. Furthermore, corporate governance practices across all countries appear to have a significant positive impact on improved financial performance, with the exception of China, where they appear to have a significant negative impact, possibly influenced by cultural values or other factors.

In the investigation of ROE results, it is worth noting that all corporate governance attributes and control variables have a positive and statistically significant relationship with the firm's financial success, highlighting the critical role of effective governance in a firm's development. This means that a one-unit rise in corporate governance variables correlates to a one-unit increase in ROE based on the value. However, China has a significant negative impact, which may be related to cultural values or other contributing factors.

The examination of EPS findings shows that, with the exception of board size and board attendance, all attributes have a statistically significant positive influence on performance. Notably, there is a considerable negative influence on board attendance, implying that a large number of participants may disturb the board's decision-making process and squander valuable time and resources. Furthermore, Malaysia is the only country with a positive effect, while other Asian economies show no distinct benefits. Finally, our findings highlight the important influence of corporate governance policies on business financial success. For more accuracy of results by removing the simultaneous effect, we used the generalized methods of movement (GMM) using the lag difference techniques.

## 4.3 Robustness analysis: Generalized Method of Movement (GMM)

The regression analysis has some weaknesses in the econometrics analysis due to the current conditions. In the current business environment, the variables are classified as structural, which shows the dual causal relationship between each other, creating a critical point on the validity of the regression analysis. The simultaneous structuring of the different factors motivates using the simultaneous equation modeling in the endogenous variables. To diagnose the real impact of the variables on the endogenous variables, we used the generalized method of movement (GMM) with the help of the instruments. In this study, we used the difference GMM to investigate the real impact of the exogenous variable on the endogenous. The motivation behind this method is the simultaneous effect of the variables, such as ROA, which can increase CSR disclosure, and CSR disclosure, which can enhance reputations, leading to profitability and vice versa. The results of the GMM methods show that the J-Statistics value is favorable, and the probability of J-Statistics is significant at a 1% level, indicating excellent estimations of the models and reliable results for interpretation. Moreover, GMM findings in Table 5 show that the ROA and EPS lag value affect the current value with different ratios and contribute positively.

Our results show that all the results improved with greater frequency and intensity compared with the regression analysis, which removed the simultaneous effects. In all models, GMM estimations show that the intensity of the effects improved, such as GDP, FS, and ENV impact, which increased corporate financial performance. In our results, the direction of

**Table 5. Robust analysis through GMM method.**

| Variable | ROA | | ROE | | EPS | |
|---|---|---|---|---|---|---|
| | Coefficient | Prob. | Coefficient | Prob. | Coefficient | Prob. |
| ROA (-1) | 0.011 | 0.001 | - | - | - | - |
| ROE (-1) | - | - | 0.001 | 0.000 | - | - |
| EPS (-1) | - | - | - | - | 0.001 | 0.058 |
| ABC | 0.423 | 0.093 | 37.871 | 0.057 | 8.921 | 0.000 |
| ACE | 0.098 | 0.000 | 0.012 | 0.000 | 0.020 | 0.000 |
| ACI | 0.018 | 0.000 | 0.180 | 0.000 | 0.087 | 0.000 |
| AGE | 0.170 | 0.000 | 0.179 | 0.000 | 0.019 | 0.000 |
| BA | 0.231 | 0.000 | 2.331 | 0.000 | -0.976 | 0.000 |
| BS | 1.520 | 0.000 | 2.191 | 0.000 | 3.171 | 0.000 |
| CSC | 0.176 | 0.000 | 2.130 | 0.000 | -0.934 | 0.000 |
| EPU | 0.012 | 0.000 | 0.097 | 0.000 | 0.089 | 0.000 |
| FS | -0.001 | 0.000 | 0.019 | 0.000 | 0.019 | 0.000 |
| GDP | 2.721 | 0.000 | 3.912 | 0.010 | 4.752 | 0.761 |
| China | -0.241 | 0.026 | -9.918 | 0.000 | 0.117 | 0.686 |
| Malaysia | 2.059 | 0.022 | 32.746 | 0.000 | 0.351 | 0.559 |
| J-Statistics | 0.909 | | 0.721 | | 0.911 | |
| Prob(J-statistics) | 0.005 | | 0.001 | | 0.002 | |
| AR (1) | 0.048 | | 0.651 | | 0.001 | |
| AR (2) | 0.871 | | 0.981 | | 0.541 | |

Note. Findings are presented with respect to Pakistan as the country dummy.

significance is the same as that of the regression analysis, so we rely on the findings of the GMM methods because the problem of autocorrelation in the regression analysis has been removed. In the ROA results, the audit board committee (ABC) has no significant impact on performance, as it did not follow the accepted criteria of probability value in the regression results, by employing the GMM methods, it has a significant and positive impact on ROA. Finally, we can conclude that GMM methods improve the regression analysis findings by removing the dual causal relationship and representing the real impact only.

## 4.4 Effect of the moderator variable

The R-squared values for our models, namely return on assets (ROA), return on equity (ROE), and earnings per share (EPS), were calculated as 0.449, 0.439, and 0.129, respectively, when the moderating effects of economic policy uncertainty were examined, as shown in Table 6. These statistics show that the explanatory variable of corporate governance has a considerable influence on the dependent variables, indicating that the model is fit for purpose. Furthermore, all calculated models had positive F-statistic values, confirming the accuracy of the estimates. The Durbin-Watson values were likewise favorable, close to 2.00, indicating that the calculated models had favorable autocorrelation values.

In Table 6, it was found that the results of Economic policy uncertainty were negative and significant moderating impact between corporate governance practices and corporate financial performance. This means that countries with higher economic policy uncertainty tend to have lower profitability in terms of ROA, ROE, and EPS. Moreover, the findings show that economic policy uncertainty has a notable and statistically significant moderating effect on the connection between the audit committee and firm financial performance. The negative

**Table 6. Results of moderation economic policy uncertainty.**

| Variable | ROA | | ROE | | EPS | |
|---|---|---|---|---|---|---|
| | Coefficient | Prob. | Coefficient | Prob. | Coefficient | Prob. |
| ABC*EPU | -0.008 | 0.000 | -0.231 | 0.006 | -0.013 | 0.001 |
| ACE*EPU | -0.013 | 0.007 | -0.000 | 0.004 | -0.001 | 0.007 |
| ACI*EPU | -0.135 | 0.002 | -0.006 | 0.006 | -0.001 | 0.000 |
| BA*EPU | -0.001 | 0.000 | -0.178 | 0.007 | -0.004 | 0.001 |
| BS*EPU | -2.132 | 0.000 | -2.873 | 0.876 | -0.006 | 0.655 |
| China | -0.037 | 0.000 | -1.164 | 0.001 | -0.198 | 0.342 |
| Malaysia | -0.041 | 0.000 | -1.089 | 0.005 | -0.766 | 0.000 |
| R-Value | 0.449 | | 0.431 | | 0.129 | |
| Adjusted R | 0.416 | | 0.421 | | 0.119 | |
| Probability | 0.000 | | 0.000 | | 0.000 | |
| Durbin-Watson | 1.991 | | 2.001 | | 2.101 | |

indication indicates that the audit committee's influence on financial performance is less beneficial in the context of significant economic policy uncertainty. This means that the audit committee's positive impact on financial performance may be diminished during economic insecurity and unpredictability periods. Firms with competent audit committees are likelier to ensure openness, risk management, and accountability. When economic policy uncertainty looms, organizations may have heightened problems in managing economic conditions and maintaining financial performance, even if they have a strong audit committee.

Similarly, the study also finds that economic policy uncertainty significantly moderates the connection between audit committee independence and firm financial performance. This shows that when economic policy uncertainty is high, the favorable impact of an independent audit committee on financial performance is lessened. These findings highlight the necessity of considering external economic conditions when evaluating the influence of corporate governance policies on financial outcomes.

However, the study finds that economic policy uncertainty significantly moderates the connection between board size and firm financial performance. This means that the positive benefit of a larger board on financial performance lessens during periods of high economic policy uncertainty. These findings emphasize the dynamic interaction of external economic variables and corporate governance dynamics in affecting financial outcomes.

The study finds that economic policy uncertainty has a significant negative moderating influence on the relationship between board attendance and business financial success. This shows that when economic policy uncertainty is high, the positive impact of board participation on financial performance diminishes. These findings highlight the contextual aspect of the relationship between corporate governance standards and financial outcomes, especially amid economic policy uncertainty. The effect of each economy is different due to the change of culture and other norms within the country.

## 5. Discussion

The audit board committee (ABC) has not affected the corporate financial performance due to the cultural and policy differences across the countries around the globe. Nonetheless, the independence of the audit committee demonstrates a noteworthy and positive impact on performance, aligning with prior research findings [58]. Furthermore, all other dimensions of corporate governance exhibit significant and positive effects, underscoring the potential of sound governance to enhance financial performance. Notably, when examining the impact

across various countries, it becomes evident that only China exhibits a significant negative effect of corporate governance on firm performance, whereas all other Asian economies display a positive effect consistent with the composite effect observed in the overall analysis. These findings hold relevance for companies operating in diverse countries as they contemplate efforts to enhance their corporate governance practices. Although good governance is commonly linked to improved financial performance, it is crucial to consider the unique country-specific context and potential disparities in the influence of distinct governance practices [59].

In this regard, our findings align with previous literature, reinforcing the idea that effective corporate governance can positively impact corporate financial performance [60]. Notably, several key attributes of corporate governance have been linked to enhanced performance, including board independence, corroborating findings similar to those of [59], who established that companies with independent boards tend to exhibit superior financial performance due to the objective oversight provided by independent directors, which helps mitigate conflicts of interest. Overall, our results are consistent with the broader body of literature suggesting that strong corporate governance practices play a pivotal role in bolstering corporate financial performance, and firms that prioritize these practices are poised for long-term success [61]. In our findings, the significant moderating impact of Economic Policy Uncertainty on the relationship between corporate governance and financial performance is in line with the observations of [62], who found that companies facing high economic uncertainty experienced decreased stock returns.

Economic policy uncertainty can harm performance across various domains, including financial markets and business operations. High levels of economic policy uncertainty often led to increased caution and risk aversion among investors and firms. Businesses may postpone investments, delay strategic decisions, and prioritize liquidity management to hedge against potential adverse economic developments. Uncertainty can also contribute to elevated financing costs, reducing firms' access to capital and potentially impacting their ability to grow and innovate. In financial markets, heightened economic policy uncertainty can increase market volatility and reduce investor confidence, resulting in decreased stock prices and returns. In sum, economic policy uncertainty can cast a shadow of apprehension, impacting performance by hindering investment, increasing costs, and generating a climate of unpredictability that can undermine corporate and financial sector performance.

## 6. Conclusion

Our study concluded that all corporate governance attributes increase financial performance, and economic policy uncertainty provides a significant negative moderating impact between them. Moreover, the effect is different in each economy due to its corporate governance mechanism and control of the corporate sector by the government. The study has several policy implications for corporations and policymakers. The government should make strong governance frameworks to reduce uncertainty and improve financial performance within the economy. Moreover, this study provides policy implications for governments to implement policies encouraging openness, accountability, and regulatory stability to boost investor trust and attract investment. Investor confidence in the organization is necessary. So, Investors should prioritize due diligence on corporate governance procedures when making investment decisions, considering the influence on financial returns. Rules of law and its legislation are important to the governance mechanism. Using this study's findings, legislators are encouraged to create legislation that enforces robust governance standards and provides efficient dispute resolution processes, fostering a favorable climate for the region's long-term economic growth and development.

This study has some limitations and provides a significant path for future research. Due to data limitations, we selected only 400 non-financial firms' data. Future research may be conducted on large data sets of 50 years or more to cover this topic. Moreover, due to the limitations of the data, we used regression analysis and GMM methodology to estimate the results. Still, future research may use the latest methodology, such as wavelet transformation by cross wavelet and wavelet coherence, to assess the relationship between these variables. Wavelet transformation is useful for removing data biases.

Additionally, it is crucial to recognize that economic policy uncertainty scores may differ between emerging economies and Asian economies, highlighting the need for future research that focuses on emerging economies and facilitates comparisons between governance practices in developed and emerging economies. Although this study encompasses non-financial firms, future research should investigate financial firms' data. Furthermore, exploring the direct relationship between economic policy uncertainty and corporate financial performance presents another bright path for future investigations.

## Supporting information

**S1 Data.**
(XLSX)

**S2 Data.**
(XLSX)

## Author Contributions

**Conceptualization:** Chaoyang Zhang.

**Project administration:** Chaoyang Zhang.

**Writing – original draft:** Chaoyang Zhang.

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
