## [Decision Letter · Decision Letter 0]

26 Feb 2024

PONE-D-23-36973Corporate Governance, Financial Performance and Economic Policy Uncertainty: Evidence From Emerging Asian EconomiesPLOS ONE

Dear Dr. Zhang,

Thank you for submitting your manuscript to PLOS ONE. After careful consideration, we feel that it has merit but does not fully meet PLOS ONE’s publication criteria as it currently stands. Therefore, we invite you to submit a revised version of the manuscript that addresses the points raised during the review process.

Do the necessary corrections based on the reports form the reviewers. Do send the paper for proofreading before resubmitting back to the journal. 

We look forward to receiving your revised manuscript.

Kind regards,

Evan Poh Hock Lau

Academic Editor

PLOS ONE

2. PLOS requires an ORCID iD for the corresponding author in Editorial Manager on papers submitted after December 6th, 2016. Please ensure that you have an ORCID iD and that it is validated in Editorial Manager. To do this, go to ‘Update my Information’ (in the upper left-hand corner of the main menu), and click on the Fetch/Validate link next to the ORCID field. This will take you to the ORCID site and allow you to create a new iD or authenticate a pre-existing iD in Editorial Manager. Please see the following video for instructions on linking an ORCID iD to your Editorial Manager account: https://www.youtube.com/watch?v=_xcclfuvtxQ".

4. We suggest you thoroughly copyedit your manuscript for language usage, spelling, and grammar. If you do not know anyone who can help you do this, you may wish to consider employing a professional scientific editing service. 

A clean copy of the edited manuscript (uploaded as the new *manuscript* file)”.

5. We note that your Data Availability Statement is currently as follows: [All relevant data are within the manuscript.Add Data Availability statement here]

6. Please amend either the abstract on the online submission form (via Edit Submission) or the abstract in the manuscript so that they are identical.

Reviewers' comments:

Reviewer's Responses to Questions

**Comments to the Author**

1. Is the manuscript technically sound, and do the data support the conclusions?

Reviewer #1: Yes

Reviewer #2: Yes

2. Has the statistical analysis been performed appropriately and rigorously? 

Reviewer #1: Yes

Reviewer #2: Yes

3. Have the authors made all data underlying the findings in their manuscript fully available?

Reviewer #1: Yes

Reviewer #2: No

4. Is the manuscript presented in an intelligible fashion and written in standard English?

Reviewer #1: Yes

Reviewer #2: Yes

5. Review Comments to the Author

Reviewer #1: I writed my notes on file. Its contributions to policy makers and literature is not clear and weakly. Sample is not enough. Authors comments must be depend on result of analyse. Comments are very generally. Some comments are weakly and not depent on literature

Reviewer #2: The manuscript entitled "Corporate Governance, Financial Performance and Economic Policy Uncertainty:

Evidence From Emerging Asian Economies" investigates an important subject. The following comment could enhance the level of the work:

1. The introduction could be strengthened by providing a more concise and focused explanation of the research gap and objectives.

2. Consider integrating more recent references to enhance the currency of the literature review. see for example:

Abdullah, H., & Tursoy, T. (2023). The Effect of Corporate Governance on Financial Performance: Evidence From a Shareholder-Oriented System. Interdisciplinary Journal of Management Studies (Formerly known as Iranian Journal of Management Studies), 16(1), 79-95. doi: 10.22059/ijms.2022.321510.674798

3. The theoretical framework is well-developed, drawing upon various theories such as agency theory, stewardship theory, and stakeholder theory. However, it would be beneficial to provide a clearer explanation of how these theories inform the hypotheses.

4. Ensure that each hypothesis is clearly stated and aligned with the theoretical underpinnings.

5. Provide more details on the specific variables used to measure corporate governance, financial performance, and economic policy uncertainty. what is DGP? do the authors mean GDP?

6. Clarify the rationale for selecting the generalized method of moments (GMM) for robustness analysis. How could you calculate Adjusted R square for GMM in table 4? How could the correlation between ROA and ROE be 1? see: Abdullah, H., Tursoy, T. Capital structure and firm performance: evidence of Germany under IFRS adoption. Rev Manag Sci 15, 379–398 (2021). https://doi.org/10.1007/s11846-019-00344-5

7. Discuss any potential limitations of the methodology, such as data availability or sampling bias.

6. PLOS authors have the option to publish the peer review history of their article (what does this mean?). If published, this will include your full peer review and any attached files.

Reviewer #1: No

Reviewer #2: No

---

## [Author Response · Author response to Decision Letter 0]

18 Mar 2024

Reviewer #1:

I writed my notes on file. Its contributions to policy makers and literature is not clear and weakly. Sample is not enough. Authors comments must be depend on result of analyse. Comments are very generally. Some comments are weak and not depent on literature.

Response to reviewer 1: Thank you for the review and suggested me your significant feedback. I corrected all the suggestions recommended through note file in the main manuscript. Please find Yellow Highlighted in manuscript for the track changes for each points. The policy implications make clearer and more innovative. The literature review are make more accurate with recent references insight. All the arguments in the results make effective based on authors own findings conclusion and its justification to previous study literature. The following correction are done according to suggestions.

1. The abbreviation of EPS properly used. Please find page 1 Abstract and line# 13 yellow highlighted.

2. The references missing in introduction cited. (Please find page 2 Line 33 and 41) 

3. The Week statement in the Introduction removed (Please find page 2 and yellow highlighted)

4. Results implications are written at page 3-line 86-93 yellow highlighted.

5. In Methodology section, unnecessary information removed. ( Please find page 8)

6. The reason for the selecting non-financial firms are given at page 8-9 yellow highlighted. Moreover, reason of selection of these economies are given.

7. GDP spelling mistakes corrected.

8. The reason for the use of GMM are given at page 12 and line #327-343.

9. The outlier in the Table 2 Descriptive statistics corrected.

10. The AR (1) and AR (2) value in GMM analysis are given at page 18 yellow heighted in the Table.

11. In Conclusion section general comments are removed.

12. The policy implication is clearly mentioned with respect to government, investors, legislators and investors. Please find page 22 and line #521-531 and yellow highlighted.

Reviewer #2:

The manuscript entitled "Corporate Governance, Financial Performance and Economic Policy Uncertainty: Evidence From Emerging Asian Economies" investigates an important subject. The following comment could enhance the level of the work:

1. The introduction could be strengthened by providing a more concise and focused explanation of the research gap and objectives.

Response to point 1: Research gap and objectives are clearly mentioned in introduction section. The background is also discussed according to the gaps and objectives. Please find page 2-3 from Line number 50-63 for track changes of the Gaps and Line # 64-71 for the Objectives.

2. Consider integrating more recent references to enhance the currency of the literature review. see for example:

Abdullah, H., & Tursoy, T. (2023). The Effect of Corporate Governance on Financial Performance: Evidence From a Shareholder-Oriented System. Interdisciplinary Journal of Management Studies (Formerly known as Iranian Journal of Management Studies), 16(1), 79-95. doi: 10.22059/ijms.2022.321510.674798

Response to point 2: The latest references added in the literature with critical review. Please find page 4-7 from Line# 99-214 green highlighted.

3. The theoretical framework is well-developed, drawing upon various theories such as agency theory, stewardship theory, and stakeholder theory. However, it would be beneficial to provide a clearer explanation of how these theories inform the hypotheses.

Response to point 3: The theories are discussed in relation to the hypothesis estimated. On the theoretical base we develop the hypothesis. Please find page 4 and line # 103-119 green highlighted for the track changes.

4. Ensure that each hypothesis is clearly stated and aligned with the theoretical underpinnings.

Response to point 4: The hypothesis are developed on the ground of the underpinning theories and the empirical literature. 

5. Provide more details on the specific variables used to measure corporate governance, financial performance, and economic policy uncertainty. what is DGP? do the authors mean GDP?

Response to point 5: More detail on variables are added according to suggestion. Please find page 10 and green highlighted in variable measurement section for the track changes. The “GDP” corrected spelling mistake.

6. Clarify the rationale for selecting the generalized method of moments (GMM) for robustness analysis. How could you calculate Adjusted R square for GMM in table 4? How could the correlation between ROA and ROE be 1? see: Abdullah, H., Tursoy, T. Capital structure and firm performance: evidence of Germany under IFRS adoption. Rev Manag Sci 15, 379–398 (2021). https://doi.org/10.1007/s11846-019-00344-5

Response to point 6: The rationale of selecting GMM for robustness analysis are given according to suggestions (Please find page 12 and line #327-343 and green highlighted for track changes). Moreover, the table 4 represents the regression Table that includes the Adjusted R square but GMM included the AR(1) and AR(2) values in Table 5. The Outlier of the correlation resolved. 

7. Discuss any potential limitations of the methodology, such as data availability or sampling bias.

Response to point 7: The limitations make wider related to methodology, such as data availability or sampling. Please find page 22-23 and line #332-345 and green highlighted.

Thank You

---

## [Decision Letter · Decision Letter 1]

4 Sep 2024

PONE-D-23-36973R1Corporate governance, financial performance, and economic policy uncertainty. Evidence from Emerging Asian EconomiesPLOS ONE

Dear Dr. Zhang,

Thank you for submitting your manuscript to PLOS ONE. After careful consideration, we feel that it has merit but does not fully meet PLOS ONE’s publication criteria as it currently stands. Therefore, we invite you to submit a revised version of the manuscript that addresses the points raised during the review process.

We look forward to receiving your revised manuscript.

Kind regards,

Ricky Chee Jiun Chia

Academic Editor

PLOS ONE

Reviewers' comments:

Reviewer's Responses to Questions

**Comments to the Author**

1. If the authors have adequately addressed your comments raised in a previous round of review and you feel that this manuscript is now acceptable for publication, you may indicate that here to bypass the “Comments to the Author” section, enter your conflict of interest statement in the “Confidential to Editor” section, and submit your "Accept" recommendation.

Reviewer #2: (No Response)

Reviewer #3: (No Response)

2. Is the manuscript technically sound, and do the data support the conclusions?

Reviewer #2: Yes

Reviewer #3: Partly

3. Has the statistical analysis been performed appropriately and rigorously? 

Reviewer #2: Yes

Reviewer #3: No

4. Have the authors made all data underlying the findings in their manuscript fully available?

Reviewer #2: Yes

Reviewer #3: Yes

5. Is the manuscript presented in an intelligible fashion and written in standard English?

Reviewer #2: Yes

Reviewer #3: No

6. Review Comments to the Author

**Reviewer #2:** The authors followed my comments all. So, the manuscript is not publishable from my perspective. . .

**Reviewer #3:** #1: My main concern is the justification and supporting resources of choosing the three countries, Pakistan, China and Malaysia as comparison. For example Pakistan chosen for its unique governance, China for transitioning to market-orientation, and Malaysia for its robust regulatory environment. Where is the further proof or justifications for these?

#2: Accordingly, there is no discussion about the comparison between the three countries, which is not aligned to the claimed contribution.

#3: Sentences in page 3 and page 22 are repeated verbatim, regarding the contribution of the study.

#4: Referring to these sentences, the second contribution is missing. From first then after that was the third.

#5: What is CSR Suitability Committee? Are you trying to actually state 'Sustainability' instead of 'suitability'?

#6: Does the item CSR committee, covers Sustainability Committees as well? What if the committee is combined with another committee and not stand alone? Is that also included?

#7: The abbreviations used for the items in your results tables for example ACI, AGE, BS, etc were not spelled out anywhere in the text. Some were, but some not. The reader may make a guess based on the conceptual framework but that is not appropriate and may be inaccurate and subjective.

#8: Assuming that my concern in #1 and #2 is addressed, and that it turns out that the choice of the 3 countries are reasonable, justified and has significant impact, I have no further comments on the method and data analysis used.

7. PLOS authors have the option to publish the peer review history of their article (what does this mean?). If published, this will include your full peer review and any attached files.

Reviewer #2: **Yes: **Hariem Abdullah

Reviewer #3: No

---

## [Author Response · Author response to Decision Letter 1]

12 Sep 2024

Reviewer #2

 The authors followed my comments all. So, the manuscript is not publishable from my perspective.

Response to Reviewer 2: Thank you for the review. In the last version, I correct all your suggestions. 

Reviewer #3

#1: My main concern is the justification and supporting resources of choosing the three countries, Pakistan, China and Malaysia as comparison. For example Pakistan chosen for its unique governance, China for transitioning to market-orientation, and Malaysia for its robust regulatory environment. Where is the further proof or justifications for these?

Response to Reviewer Point 1:

The proper justification are given in introduction section and also in methodology section according to the suggestion. Please find page 3 and yellow highlighted as “Pakistan's relatively nascent corporate governance framework often struggles with enforcement, which hinder financial performance, particularly during periods of high economic policy uncertainty (Shaikh, Khoso, & Jummani, 2024). Similarly, China with more centralized governance structures and state-owned enterprises, demonstrates a different dynamic where government policies and interventions mitigate some uncertainty but may also stifle market-driven growth (Cao, Prior, Gu, & Giurco, 2023). In contrast, Malaysia exhibits a more developed corporate governance system that fosters better financial performance in the face of uncertainty, with robust regulatory frameworks allowing for greater market resilience (Y. Wu & Tham, 2023). The influence of economic policy uncertainty in each country is shaped by the strength of institutions and governance practices, impacting corporate outcomes differently. Therefore, this study extracts a gap in literature to investigate the corporate governance along with the economic policy uncertainty effect the financial performance in different culture that provide strong evidence of both strong and week corporate governance structure. Similarly, this study is also important for improving the corporate sector in the emerging economy where corporate governance mechanism is week.”

#2: Accordingly, there is no discussion about the comparison between the three countries, which is not aligned to the claimed contribution.

Response to Reviewer Point 2: The discussion on comparison are written according to suggestions with references, Please find page 3 yellow highlighted and page 8-9 yellow highlighted for track changes.

#3: Sentences in page 3 and page 22 are repeated verbatim, regarding the contribution of the study.

Response to Reviewer Point 3: The repeating words deleted. 

#4: Referring to these sentences, the second contribution is missing. From first then after that was the third.

Response to Reviewer Point 4: In contribution paragraph, the refered objective corrected. Please find page 3 yellow highlighted for track changes.

#5: What is CSR Suitability Committee? Are you trying to actually state 'Sustainability' instead of 'suitability'?

Response to Reviewer Point 5: It is CSR committee existence that corrected according to suggestions in the manuscript. 

#6: Does the item CSR committee, covers Sustainability Committees as well? What if the committee is combined with another committee and not stand alone? Is that also included?

Response to Reviewer Point 6: It is CSR committee existence that corrected according to suggestions in the manuscript.

#7: The abbreviations used for the items in your results tables for example ACI, AGE, BS, etc were not spelled out anywhere in the text. Some were, but some not. The reader may make a guess based on the conceptual framework but that is not appropriate and may be inaccurate and subjective.

Response to Reviewer Point 7: Abbreviation make corrected according to suggestions. Please find page 11 yellow highlighted and also under the table of descriptive statistics for first uses.

#8: Assuming that my concern in #1 and #2 is addressed, and that it turns out that the choice of the 3 countries are reasonable, justified and has significant impact, I have no further comments on the method and data analysis used.

Response to Reviewer Point 8: The selections of three countries are given in Introduction and also in methodology.

---

## [Editor Report · Decision Letter 2]

4 Oct 2024

Corporate governance, financial performance, and economic policy uncertainty. Evidence from Emerging Asian Economies

PONE-D-23-36973R2

Dear Dr. Chaoyang Zhang,

We’re pleased to inform you that your manuscript has been judged scientifically suitable for publication and will be formally accepted for publication once it meets all outstanding technical requirements.

Kind regards,

Ricky Chee Jiun Chia

Academic Editor

PLOS ONE
---

## [Editor Report · Acceptance letter]

9 Oct 2024

PONE-D-23-36973R2 

PLOS ONE

Dear Dr. Zhang, 

I'm pleased to inform you that your manuscript has been deemed suitable for publication in PLOS ONE. Congratulations! Your manuscript is now being handed over to our production team.

Kind regards, 

on behalf of

Dr. Ricky Chee Jiun Chia 

Academic Editor

PLOS ONE